# A Prospective Study of Etiology and Auditory Profiles in Infants with Congenital Unilateral Sensorineural Hearing Loss

**DOI:** 10.3390/jcm11143966

**Published:** 2022-07-07

**Authors:** Marlin Johansson, Eva Karltorp, Kaijsa Edholm, Maria Drott, Erik Berninger

**Affiliations:** 1Division of Ear, Nose and Throat Diseases, Department of Clinical Science, Intervention and Technology, Karolinska Institutet, 141 52 Stockholm, Sweden; eva.karltorp@regionstockholm.se (E.K.); kaijsa.edholm@regionstockholm.se (K.E.); erik.berninger@ki.se (E.B.); 2Department of Audiology and Neurotology, Karolinska University Hospital, 141 86 Stockholm, Sweden; maria.drott@regionstockholm.se; 3Department of Hearing Implants, Karolinska University Hospital, 141 86 Stockholm, Sweden; 4Department of Radiology, Karolinska University Hospital, 141 57 Stockholm, Sweden

**Keywords:** etiology, sensorineural hearing loss, unilateral hearing loss, imaging, cytomegalovirus, electrophysiology, pediatric, congenital, auditory dysfunction

## Abstract

Congenital unilateral sensorineural hearing loss (uSNHL) is associated with speech-language delays and academic difficulties. Yet, controversy exists in the choice of diagnosis and intervention methods. A cross-sectional prospective design was used to study hearing loss cause in twenty infants with congenital uSNHL consecutively recruited from a universal neonatal hearing-screening program. All normal-hearing ears showed ≤20 dB nHL auditory brainstem response (ABR) thresholds (ABRthrs). The impaired ear median ABRthr was 55 dB nHL, where 40% had no recordable ABRthr. None of the subjects tested positive for congenital cytomegalovirus (CMV) infection. Fourteen subjects agreed to participate in magnetic resonance imaging (MRI). Malformations were common for all degrees of uSNHL and found in 64% of all scans. Half of the MRIs demonstrated cochlear nerve aplasia or severe hypoplasia and 29% showed inner ear malformations. Impaired ear and normal-hearing ear ABR input/output functions on a group level for subjects with ABRthrs < 90 dB nHL were parallel shifted. A significant difference in interaural acoustic reflex thresholds (ARTs) existed. In congenital uSNHL, MRI is powerful in finding a possible hearing loss cause, while congenital CMV infection may be relatively uncommon. ABRs and ARTs indicated an absence of loudness recruitment, with implications for further research on hearing devices.

## 1. Introduction

Congenital unilateral sensorineural hearing loss (uSNHL) is a chronic condition affecting approximately 1 in every 2000 newborns [1,2]. Children with uSNHL have a higher risk of speech-language delays, academic difficulties, and psychosocial challenges compared to children with normal hearing (e.g., [3,4,5]). They generally also have impaired binaural hearing, including sound source localization [6,7,8,9], and impaired speech recognition in noise [9,10,11].

Controversy exists in the choice of diagnostic methods for children with uSNHL. For example, some argue for using imaging [12,13,14], while a recent systematic review and meta-analysis argues that “imaging does not provide information that alters management or gives insight into prognosis or hereditary causes in most cases and therefore does not warrant strong recommendations” [15]. The prevalence of hearing loss causes may be different for congenital uSNHL, compared to uSNHL that emerges later in development. A systematic investigation should be of value for clinical routines, as congenital uSNHL represents one fourth of the sensorineural hearing losses found through the universal neonatal hearing-screening programs [2].

Malformations of the auditory system are more common in congenital uSNHL compared to bilateral SNHL [16,17]. The malformation prevalence depends on several factors, e.g., the use of magnetic resonance imaging (MRI) and/or computed tomography (CT). CT provides a good view of the bony labyrinth and middle ear, whereas soft-tissue resolution is higher with modern MRI and is therefore the preferred method for evaluating the vestibulocochlear nerve with its cochlear branch, the facial nerve, and the internal cochlear structures. The malformation prevalence with MRI is about 37–58% in children with uSNHL [12,13,14,18,19], and similarly around 36–67% for CT [13,20,21]. The prevalence may be somewhat higher, around 46–67%, when obviously acquired uSNHL has been excluded, e.g., meningitis or trauma [12,20,21]. However, more precise estimates of malformation prevalence for congenital uSNHL are lacking.

The congenital cytomegalovirus (cCMV) infection is believed to be the leading non-genetic cause for SNHL during the last few decades, with a prevalence of up to 20–30% in SNHL [22,23,24]. It is common that SNHL associated with cCMV infection is progressive and sometimes even fluctuating [24,25,26]. The debut of SNHL may even be more common after the neonatal period, so that many children with cCMV-associated SNHL will not be identified through universal neonatal hearing-screening (UNHS) programs [27,28]. Children with uSNHL (congenital and acquired) have previously been found to be cCMV positive in around 10% [14] to 20% of cases [29,30]. Still, the prevalence of uSNHL caused by cCMV infection needs more research, due to, e.g., small groups of children, the frequent uncertainty of uSNHL onset when not found through UNHS, and the variability in spread of cCMV infection over time and across countries [14,29,30].

Research of auditory profiles or hearing function in relation to etiology is lacking. Most research on etiology and auditory profiles in uSNHL focuses on cochlear implant (CI) candidates with profound uSNHL, i.e., single-sided deafness (SSD) (e.g., [30,31]), and is much less explored in mild-moderate uSNHL, i.e., potential hearing aid (HA) candidates. All patients with hearing loss experience elevated hearing thresholds in at least one ear, but other auditory functions may vary between patients. Loudness recruitment, i.e., an increased growth in perceived loudness as the intensity level of a signal increases, is an important amplification factor in a hearing aid, that most adult patients with cochlear hearing loss experience [32,33]. The prevalence of loudness recruitment in the pediatric population is sparsely explored, as young children are difficult to assess with psychoacoustic tests of loudness perception. The auditory brainstem response (ABR) may be used to estimate loudness perception by neural firing [34,35] and needs further study. Congenital uSNHL represents an excellent model for estimation due to the possibility of comparing the impaired ear (IE) with the normal-hearing ear (NE).

The aim was to describe congenital uSNHL (ranging from mild to profound) in infants, focusing on the causes, hearing function and the affected auditory mechanisms by using a prospective study design and a comprehensive test battery. Tests included MRI, cCMV infection testing, acoustic reflex threshold (ART) measurements to investigate middle ear, inner ear and neural function (e.g., [36,37]), and various otoacoustic emissions (OAEs) to investigate cochlear function [38,39]. The auditory brainstem response (ABR) was measured for site of lesion testing, hearing threshold estimation, and to study the function of the cochlea and auditory pathways (e.g., [40,41,42]). The ABR amplitude as a function of stimulus level, i.e., the input/output (I/O) function, was studied to estimate loudness recruitment.

## 2. Materials and Methods

### 2.1. Study Design

The subjects were recruited during the years 2019–2020 from the bedside UNHS program in Region Stockholm, which is based on multiple transient-evoked OAE (TEOAE) recordings and an automatic ABR (aABR). The UNHS program has been described previously (first TEOAE around postnatal day 3, 98% screened) [2,42,43,44], where the aABR screening before clinical ABR was first introduced in 2016 [45].

Eligible subjects were invited to participate in two research visits. ABRs, TEOAEs, distortion-product OAEs (DPOAEs), tympanograms, and ARTs were recorded at the first visit. The ABRs, TEOAEs and DPOAEs were recorded in an audiometric test room with low ambient sound levels, allowing determinations down to −10 dB HL (ISO8253-1 2010). The tympanograms and ARTs were measured in the audiometric test room or a quiet room. Otomicroscopy was performed by an experienced otologist to exclude middle ear disease. An oral case history was obtained from the infants’ parents for additional information including parents’ knowledge about relevant medical and development history, including prenatal and perinatal history. Testing for cCMV followed as soon as possible. The visit took approximately 2.5–3 h.

At a second research visit, all eligible subjects were invited to the MRI of the auditory system (inner ears and cochlear nerve).

This study was approved by the Swedish Ethical Review Authority (no: 2018/1500-31). Written informed consent was obtained from all parents with child custody.

### 2.2. Subjects

Several recruitment steps were undertaken with the aim of inviting all children born with uSNHL and excluding all children with temporary hearing loss, e.g., otitis media with effusion (OME). Inclusion criteria comprised: (1) One ear passing and one ear failing TEOAE UNHS; (2) An ABR threshold (ABRthr) > 30 dB nHL in the IE; and (3) An ABRthr of ≤20 dB nHL in the NE.

M.J. (first author) was notified when a neonate was suspected of having a unilateral hearing loss, i.e., failed several TEOAE tests in one ear and passed in the other ear, passed aABR screening in one ear (≤30 dB nHL, i.e., the lowest sound level tested) and failed in the other ear (>30 dB nHL). TEOAE pass criteria included: ≥70% whole wave reproducibility, signal-to-noise ratio (SNR) ≥4 dB in at least three of the upper four wide-frequency bands provided by the ILO instrument, and ≥50 sweeps. The neonate’s families were informed of the study and invited to the next step in the study recruitment process at the first audiologic test visit following UNHS if the ABR showed thresholds >30 dB nHL in the IE and a SNHL was alleged (based on the ABR, additional auditory steady-state response (ASSR), tympanograms, TEOAEs, and sometimes bone-conduction ABR), and ≤25 dB nHL in the NE.

In total, 48 out of 68 potential subjects based on the UNHS were excluded at the first audiologic test visit following screening or, in a few cases, at the first research visit as they did not fulfill the study inclusion criteria. This was typically due to OME that fully explained the elevated ABRthr in the IE, or an ABRthr ≤ 30 dB nHL in the IE, or bilateral hearing loss (≥25 dB nHL in both ears).

Thus, 20 infants with congenital uSNHL were invited to the study, and the 20 families all consented to participate in the study (see Figure 1 for sample size compared to estimated sample).

### 2.3. Auditory Brainstem Response (ABR)

The ABR was recorded monaurally in both ears using 100 µs rarefaction clicks at a repetition rate of 39 Hz. Clicks were presented through insert earphones (EAR Tone; Etymotic Research Inc, Elk Grove Village, IL, USA) with Eclipse EP25 (program version 4.3.0.17, Interacoustics, Middelfart, Denmark).

The ABR was recorded in 10 dB steps whenever possible, from 70 dB nHL down to the threshold or 20 dB nHL minimum. The 70 dB nHL level was chosen as the main supra-threshold level for analysis of the ABR I–V interval. If the ABR waves I and V could not be discerned at 70 dB nHL, the input level was increased in 10 dB steps up to a maximum of 80 dB nHL for the NE, and 90 dB nHL for the IE.

The response quality was enhanced by the use of up to 10,000 sweeps at stimulus levels close to the ABRthr, and ≈2000 sweeps for the other stimulus levels. Calibration of 100 µsec clicks was performed according to ISO389-6 (2007) in an occluded ear simulator IEC60318-4 (2010) [47,48].

Ag/AgCl electrodes were placed on left and right mastoids (left, right), and the forehead (vertex, ground) for differential recording between ipsilateral mastoid (noninverting) and the forehead (inverting). The subject was placed in a supine position in the parent’s arms or in a baby carrier. Lights in the audiometric test room were turned off during the ABR recordings in order to minimize electric interferences.

E.B. determined all the ABR waves, blinded to which subject and ear that was evaluated. The ABR latencies were determined using narrow-band filtering (digital filter, 300 to 1500 Hz), thereby obtaining a well-defined vertex-positive peak.

If no ABR-waves could be determined, the wave reproducibility (*ρ*) was objectively determined in the time domain (1–15 ms) for the entire ABR [42], and the lack of a response was confirmed if *ρ* was <70% (*ρ* = 70% corresponds to SNR = 3.7 dB [42]). The wave I of each ABR recording was objectively confirmed by a *ρ* ≥ 70% within a time window of 1–1.5 ms encompassing the wave. The ABR wave V at the threshold was confirmed with a *ρ* ≥ 70% within a time window of 1–1.5 ms encompassing the wave. The amplitudes for the ABR waves were quantified based on the difference between the vertex positive peak maximum and succeeding minimum [42].

Contralateral masking was applied when needed [49].

### 2.4. Transient-Evoked Otoacoustic Emission (TEOAE)

The TEOAEs were recorded in the non-linear quickscreen neonate diagnostic mode with Echoport ILO288 USB-II (Otodynamics Ltd., Hatfield, UK, program version 6, time window: 3.0–12.8 ms, click rate 78 Hz). For details on the non-linear stimulus paradigm see [50]. The Echoport interface was placed in the audiometric test room during the measurement together with the subject and accompanying parent, whereas the tester and computer were outside in the surrounding quiet room. All the TEOAE stimulus and response levels were recorded at the probe-tip in the outer ear canal using an electrically constant stimulus, corresponding to a typical stimulus level of 81.5–82 dB SPL peak [43,44]. The NE was tested first in 65% of subjects, and the right ear first in 60% of subjects, which may be important, as TEOAEs have been demonstrated to be larger in the ear tested first [51]. A sweep consisted of two sets of non-linear responses in response to four 80-μs click stimulations. Whole-wave reproducibility was calculated as the correlation coefficient of interleaved non-linear responses.

### 2.5. Distortion Product Otoacoustic Emission (DPOAE)

The DPOAEs were recorded using the same test setup as the TEOAEs with Echoport ILO288 USB-II (Otodynamics Ltd., Hatfield, UK, program version 6). The 2f_1_-f_2_ cubic distortion product component was measured with a frequency ratio of the primaries of 1.22 (f_2_/f_1_). The stimulus consisted of two equal-level sinusoids with an expected sound pressure level of 75 dB SPL at the tympanic membrane (as used by, e.g., [52]). The DPOAEs were recorded with f_2_ at 1, 1.5, 2, 3, 4, 6 and 8 kHz. The SNR was defined as the DPOAE level–(Noise + 2 SD) in dB. The NE was tested first in 63% of subjects, and the right ear first in 53% of subjects.

### 2.6. Tympanometry and Acoustic Stapedius Reflex Threshold (ART)

Tympanograms and ARTs were recorded in both ears with GSI Tympstar (Grason-Stadler, Eden Prairie, MN, USA) to primarily study the middle ear system and the function of the reflex arc. The ART was elicited ipsilaterally at tympanic peak pressure and at the stimulus frequency of 1000 Hz, using an ascending method in steps of 5 dB with a maximum of 105 dB HL (as recorded in a 2-cc coupler). The criterion for a reflex response was a repeatable compliance change corresponding to 0.02 mL or greater, supported with an impedance growth at a higher stimulus level.

### 2.7. Magnetic Resonance Imaging (MRI)

Subjects were excluded from MRI if they were too ill for sedation with dexmedetomidine, which was determined in two steps, first by the responsible otologist (E.K.), then by the MRI team physician. One subject was excluded from MRI due to a congenital heart abnormality. Another 5 out of 19 subjects declined MRI participation (Figure 1).

All but one of the 14 MRI scans were assessed with 3T scanners (Siemens Skyra, Erlangen, Germany, *n* = 8, or Siemens Prisma, Erlangen, Germany, *n* = 5), the exception being for S3, where a 1.5T scanner was used (GE Optima, GE Healthcare, Fairfield, CT, USA), as a back and spine MRI was prioritized additional to the auditory system MRI.

For all 3T scans, a protocol designed for pre-cochlear implantation was used. For assessment of the inner ear, a transversal high resolution T2 3D SPACE sequence with 0.5 mm slice thickness was used. The cochlear nerve was assessed using a T2 3D SPACE sequence acquired in an oblique sagittal plane perpendicular to the inner auditory canal. The protocol also included T1-weighted images (T1WI) and T2-weighted images (T2WI) of the temporal bone, as well as sagittal T1WI, transversal T2-weighted Fluid Attenuated Inversion Recovery images (T2 FLAIR), diffusion-weighted imaging (DWI) and susceptibility-weighted imaging (SWI) of the brain.

The 1.5T scan was performed using a protocol designed for the temporal bone. The inner ear was assessed using a transversal T2 3D FIESTA sequence with 0.6 mm thickness, and the cochlear nerve by making oblique sagittal reconstructions from this sequence. The scan also included sagittal and coronal T2WI as well as transversal T1WI, T2 FLAIR and DWI of the brain.

The MRI results were reviewed by one or two experienced neuroradiologists or head-neck radiologists, and later all results were double checked by K.E., an experienced head-neck radiologist.

### 2.8. Congenital Cytomegalovirus (cCMV) Infection

A blood sample was taken from the mother for CMV testing as soon as possible after the study inclusion. If the mother had a positive CMV-test (lgG and/or lgM antibodies) the child’s newborn dried blood spot (DBS) card was then analyzed for CMV DNA with the polymerase chain reaction (PCR) technique. Exceptions were made from the study protocol for two subjects where the DBS cards were analyzed directly, as the mothers had not yet taken the blood sample after several months, despite repeated reminders, and one subject had already been CMV tested with a blood sample the same day as birth.

### 2.9. Statistical Analysis

All the statistical analyses were performed with Statistica version 13.5 (TIBCO software Inc., Palo Alto, CA, USA), except linear mixed modelling, which was performed with R version 3.4.2 (R Foundation for Statistical Computing, Vienna, Austria).

Mean and SD are primarily presented in this study for easy comparison with previous studies, although medians and interquartile ranges (IQRs) are presented when Kurtosis and Skewness were clearly different from zero, e.g., for ARTs and ABRthrs.

Nonparametric tests were used for statistical testing, due to small sample sizes.

Due to the parallel-shifted IE and NE ABR I/O functions on the group level, a post hoc analysis with linear mixed modelling of ABR amplitude was performed (waves I, III, and V, respectively), with the ear (NE/IE) and stimulus level as fixed effects and the subject as the random variable (Satterwaite’s method for *t*-tests).

## 3. Results

### 3.1. Subjects

Twenty subjects with mild to profound congenital uSNHL participated in the study (50% males, 65% left IEs, 55% first-borns) (Table 1). Subjects 15 and 17 (S15 and S17) were born as dizygotic twins (not co-twins). Six subjects spent several days in the neonatal intensive care unit (NICU) after birth due to asphyxia, anal atresia, corpus callosum agenesis, jaundice, mild respiratory distress syndrome or heart abnormality (S1, S3, S10, S12, S14 and S19). S12 was born preterm before 37 weeks of gestation (36 weeks + 1 day). All other 19 subjects were born full-term (weeks 37 to 42).

The median chronological age at the first research visit, and the diagnosis of congenital uSNHL, was 2.2 months (Table 1). The otomicroscopic examinations were normal. However, S8 showed signs of OME when first scheduled for the visit (supported by a flat tympanogram and lack of TEOAEs in the NE) so the visit was rescheduled, hence the comparably late diagnosis (Table 1). S13 was also diagnosed with congenital uSNHL comparably late, due to COVID-19 outbreak during spring 2020 and disease risk groups in the family (Table 1).

### 3.2. Auditory Brainstem Response (ABR)

All subjects slept during the ABR recording. The maximum electrode impedances (33 Hz square wave) were low with a median of 1.5 kΩ (IQR = 1–2 kΩ, *n* = 20 subjects).

#### 3.2.1. ABR Thresholds (ABRthrs)

ABRthrs of 20 dB nHL were recorded in all NEs (*n* = 20), with a wave-V reproducibility of >73% within a time window of 1–1.5 ms encompassing the wave (median *ρ* = 97%).

The median ABRthr was 55 dB nHL in the IE (IQR = 40 dB nHL-no response, *n* = 20). In 8 subjects (40%), no ABR waves could be discerned in the IE at 90 dB nHL (Table 1), indicating SSD (objectively supported by *ρ* < 70%, i.e., SNR < 3.7 dB, for an analysis window of 1–15 ms [42]). For the subjects with discernible ABR waves, the median ABRthr was 43 dB nHL (IQR = 40–48 dB nHL, *n* = 12).

#### 3.2.2. ABR I–V Intervals

The ABR interpeak I–V interval was used to determine the neural transmission time between the cochlea and upper brainstem. No statistically significant interaural difference existed in the ABR I–V wave interval (Wilcoxon matched pairs; *p* = 0.26, *n* = 9). However, the interaural between-subject variability was large (Table 1). At 70 dB nHL, the mean ± SD I–V interval was 4.80 ± 0.34 ms in the NE (*n* = 17) and 4.92 ± 0.34 ms in the IE (*n* = 8). At 80 dB nHL, it was 4.82 ± 0.31 ms in the NE (*n* = 8) and 4.81 ± 0.19 ms in the IE (*n* = 7).

#### 3.2.3. ABR Latencies

Not surprisingly, the latency as a function of the stimulus level for waves I, III and V was similar when comparing the NE and IE for all the subjects (Figure 2) (Mann–Whitney U test: *p*s > 0.05). A slight, albeit significant, interaural wave-V latency difference existed for paired comparisons at 70 dB nHL (median difference 0.25 ms, *p* = 0.03, *n* = 12), and at 80 dB nHL (median difference 0.13 ms, *p* = 0.04, *n* = 5), but not at 40, 50 or 60 dB nHL (Wilcoxon’s matched pairs test, see Appendix A). No statistically significant interaural difference existed in ABR latency for wave I and III at any stimulus level, neither paired nor unpaired (see Appendix A, *p*s > 0.05).

#### 3.2.4. ABR Wave Amplitudes

The NE vs. IE I/O functions for waves I, III and V were parallel shifted (Figure 3).

Likewise, linear mixed modelling revealed a significant effect of ear (NE vs. IE) on wave V (r_totalmodel_ = 0.78, *p* < 0.001), wave-III (r_total model_ = 0.76, *p* < 0.001) and wave I (r_total model_ = 0.64, *p* < 0.001) amplitudes. The effect of the stimulus level was also significant (*p* < 0.001, *p* < 0.001, *p* = 0.004) for wave V, III, and I, respectively. Subject was a significant contributor to the model for wave III (*p* < 0.001) and I (*p* < 0.001). 

For detailed ABR mean and median amplitude NE vs. IE difference by stimulus level with statistical significance (paired and unpaired) for wave I, III and V see Appendix A.

### 3.3. Magnetic Resonance Imaging (MRI)

Seventy percent (14/20) of the subjects underwent MRI of the auditory systems (Table 1). One subject was excluded due to congenital heart abnormality, and another 5 subjects declined MRI participation for personal reasons. The median age for MRI was 8 months (IQR = 7–10 months).

Malformations of the auditory system were identified in 64% (9/14) of the subjects. The prevalence of malformations was higher in children with profound uSNHL, as 86% (6/7) of the children with an ABRthr >90 dB nHL showed malformations, whereas children with mild to profound uSNHL demonstrated a malformation prevalence of 43% (3/7; Table 1).

Half of the MRIs revealed no visible cochlear nerve (Figure 4) (*n* = 7/14; S3, S6, S8, S11–12, S14 and S17), henceforth referred to as cochlear nerve aplasia, although severe hypoplasia cannot be excluded. All cochlear nerve aplasia was combined with hypoplasia of the vestibulocochlear nerve. Six of these seven subjects demonstrated no ABR response (or threshold >90 dB nHL, i.e., SSD), while the ABRthr for S17 was 60 dB nHL (Table 1). Moreover, no visible cochlear nerve was combined with hypoplasia of the inner ear canal in three of the subjects (S8, S11–12; i.e., inner ear canal being evidently smaller on IE side compared to NE side).

Cochlear nerve aplasia in combination with an inner ear malformation was observed in two subjects: S6 with cochlear aplasia, labyrinth dysplasia and semicircular canal dysplasia, and S3 with cochlear hypoplasia, semicircular canal dysplasia and enlarged vestibular aqueduct (EVA). The total percentage of inner ear malformations was 29% (S3, S6–7, S18; Table 1). S3, S7 and S18 revealed EVA, bilateral for S7 (ABRthr of 45 dB nHL in the IE) and unilateral for S3 (SSD), as well as for S18 (ABRthr of 60 dB nHL in the IE). S7 progressed to a bilateral hearing loss at a hearing follow-up at 8 months of age, despite having an ABRthr of 20 dB nHL, as well as TEOAEs and DPOAEs, in the NE at 2 months of age.

Nothing abnormal was identified in the auditory pathway for S2, S5, S13, S15 and S16.

### 3.4. Congenital Cytomegalovirus (cCMV) Infection

All subjects were CMV negative according to the PCR analysis on the DSP cards typically taken 48 h after birth (*n* = 16), plasma test on the same day as birth (*n* = 1), or the mother’s lgG and lgM negative CMV blood test shortly after birth (*n* = 3). The mothers who were CMV lgG and lgM negative left the blood sample for the analysis within 3 months after childbirth (51–88 days).

### 3.5. Tympanometry and Acoustic Stapedius Reflex Threshold (ART)

The change in absolute compliance, measured in equivalent volume of air at tympanic peak pressure was almost identical in the two ears, with medians of 1 mL in the NE (IQR: 0.5–1.5 mL, *n* = 19) and 1 mL in the IE (IQR: 0.7–1.4 mL, *n* = 16). Similarly, the median tympanic peak pressure was 10 daPa in the NE (IQR: −5–40 daPa, *n* = 19) and 10 daPa and IE (IQR: −10–30 daPa, *n* = 16). The ears that could not be tested at the research visit (the IEs of S1, S2, S16 and S19, and the NE of S2) showed normal tympanograms at follow-up visits (≥0.2 mL compliance, −100 daPa to 100 daPa) in combination with no significant changes in hearing (ABRthrs, ARTs and/or TEOAEs).

An ipsilateral acoustic reflex test was completed in 70% of the infants in the NEs and 70% in the IEs. The median ART was 85 dB HL in the NEs (IQR: 75–90 dB HL, *n* = 14), and 95 dB HL in the IEs (IQR: 95 dB HL-no response, *n* = 14; Table 1 for the ART of the IEs for each subject). A statistically significant median interaural ART difference of 15 dB was found for the subjects with recordable ARTs in both ears (Wilcoxons matched pairs: *p* = 0.008, *n* = 9).

### 3.6. Transient-Evoked Otoacoustic Emission (TEOAE)

All TEOAE recordings in NEs showed a SNR > 4 dB in at least three of the upper four wide-frequency bands (*n* = 18, no reliable/no recording for S2 and S10 at test, though all NEs had passed TEOAE screening previously). The mean ± SD response level was 19.8 ± 7.2 dB SPL (*n* = 18) for NEs, similar to the 19–20 dB SPL in previous research of neonatal NEs with a similar test setup [43,44]. In Nes, the median stimulus level was 81.9 dB SPL peak (IQR: 81.6–83.1, *n* = 18), the stimulus stability was high (median = 98%, IQR: 92–99, *n* = 18), and a median of 233 sweeps were used (IQR: 160–260 sweeps; *n* = 18). The whole-wave reproducibility was high with a median of 90% (IQR: 83–97%; *n* = 18), and the median entire SNR was 6.4 dB (IQR: 3.8–12.8 dB, *n* = 18). The IEs did not pass UNHS and likewise no TEOAEs could be recorded at the research visit (*n* = 18, no reliable/no recording for S10 and S19 at test, though all IEs had not passed TEOAE screening previously).

### 3.7. Distortion-Product Otoacoustic Emission (DPOAE)

A DPOAE response, classified as a positive SNR, was recorded at three frequencies or more (1–8 kHz) in 16/17 NEs (only two NE frequency responses for S19, that held the noisiest recording) and at two frequencies or more for 17/17 NEs (no reliable/no DPOAE recording for S2, S10 and S15).

In comparison, a DPOAE response was recorded at three frequencies or more (1–8 kHz) in 5/18 IEs (S2, S4, S5, S7 and S10), all with corresponding IE ABRthrs of 35–45 dB nHL (no reliable/no DPOAE recording for S13 and S15) (Table 1). DPOAE responses at two frequencies or more were recorded in 7/18 IEs (for S2–5, S7, S10 and S17) and at one frequency or more in 13/18 IEs (S1–5, S7, S9–11, S16–19).

Although DPOAEs were recorded in some IEs, the DPOAE response levels (dB SPL) were generally weaker in the IEs compared to the NEs, as expected. For the frequencies with DPOAEs in the IEs (27 frequency recordings; DPOAE level–(Noise + 2 SD) > 0 dB), the median response level was −0.2 dB SPL (IQR: −3.8 to 6.3 dB SPL), in comparison to 17.7 dB SPL (IQR: 12.5–21.7 dB SPL, 94 frequency recordings) in the NEs.

## 4. Discussion

### 4.1. Magnetic Resonance Imaging (MRI)

Infants with mild to profound congenital uSNHL, consecutively recruited from the newborn hearing-screening program, were diagnosed with malformations of the auditory system in 64% of cases (9/14); 86% for congenital SSD (6/7) and 43% for mild to severe congenital uSNHL (3/7). Thus, the diagnostic yield of 64% was numerically somewhat higher than other studies of uSNHL and MRI of 37–58% [12,13,14,18,19]. However, compared to studies where apparently acquired uSNHL had been excluded (CT and/or MRI), the 64% was very similar to the previous 46–67% malformations [12,20,21]. A strength with this study design of congenital uSNHL was that the median diagnostic age was 2 months (Table 1), in comparison to the mean/median diagnostic age of around 4 years in previous studies, where apparently acquired uSNHL had been excluded [12,20,21]. A thorough audiologic examination was also performed in addition to the MRI. In agreement with previous studies of uSNHL, malformations were common for all degrees of uSNHL [13,16], and malformations were about twice as common for children with SSD compared to children with mild to severe uSNHL [16]. For mild to severe uSNHL the 43% malformation prevalence was similar to the 44–48% [13,16]. The 86% malformation prevalence for SSD was numerically higher than the 52–71% [13,16] found previously for SSD, probably due to the high prevalence of cochlear nerve aplasia found here.

The most common malformation was cochlear nerve aplasia in 50% of scans (7/14, Figure 4), which was numerically higher than what has been found in previous studies of children with uSNHL at 17–36% [12,13,14,18,19]. Not surprisingly, cochlear nerve aplasia was much more common for SSD (6/7, 86%) than for mild-to-severe uSNHL (1/7, 14%). The proportion of subjects with SSD compared to other degrees of uSNHL does not fully explain the high prevalence of cochlear nerve aplasia in this study, as 50% of the subjects who underwent MRI had SSD, which was similar to the 28–74% SSD in previous uSNHL studies [12,13,14,18,19]. The high prevalence of cochlear nerve aplasia may be due to our aim of only including infants with congenital uSNHL, by recruiting from the UNHS program, and diagnosing the uSNHL early (median age 2 months) and performing MRI as soon as possible (median age 8 months). For instance, one study found a prevalence of cochlear nerve aplasia or hypoplasia in 100% of infants with SSD (*n* = 10) and 75% in preschool children with SSD (*n* = 20), compared to 48% in children with SSD in general (*n* = 50) [18], indicating that the diagnosis age matters. Furthermore, the MRI resolution needs to be sufficient to resolve individual nerves in the inner auditory canal and diagnose cochlear nerve aplasia or hypoplasia. A 3T MRI scanner was used in 93% of scans, the exception being for S3 where a 1.5 T scanner was used, as a back and spine MRI was prioritized additional to the auditory system MRI. Thus, the 3T scans may also contribute to the high prevalence of malformations, e.g., some studies included in a recent meta-analysis [15] found no cases of nerve aplasia or severe hypoplasia [53,54], and the MRI resolution was not described/specified, or found only a few percent cochlear nerve aplasia or severe hypoplasia [55] with MRI resolution at 1–2 Tesla.

The 29% inner ear malformations (4/14) were similar in prevalence to the 28–46% in previous studies [12,13,14,18], and so was the 21% EVA (3/14) compared to the 8–25% in previous studies [12,13,14,18]. The bilateral EVA quickly deteriorated to a bilateral SNHL, so the early MRI (at 7 months of age) contributed to a CI-fitting shortly after hearing loss onset.

### 4.2. Congenital Cytomegalovirus (cCMV) Infection

All subjects were cCMV negative. Children with uSNHL has previously been found to be cCMV positive in around 10% [14] to 20% of cases [29,30], so we expected some positive cases. The CMV PCR technique, used for the majority of subjects, has demonstrated a high negative predictive value (NPV) of 0.991 (95% CI = 0.972–0.997), as found by a meta-analysis based on 15 studies [56]. One contributing factor to the absence of cCMV infection could be that only congenital uSNHL was studied, compared to both acquired and congenital uSNHL [14,29,30]. About 30–50% children with cCMV that develop SNHL or uSNHL specifically do this during the neonatal period [27,28]. The first TEOAE measurement is also routinely performed very early, around postnatal day 3, in the UNHS program in Region Stockholm (98% screened) [2,43,44], compared to the neonatal period up to 2 months of age in previous studies [27,28]. Moreover, the spread of infections generally changes over time. The COVID-19 pandemic in 2020 to 2021 might have contributed to a lower prevalence of cCMV infection in infants, due to adjustments in hand hygiene and close contact with other people, as has been suggested from changes in cCMV prevalence in populations affected by strict lockdowns due to the pandemic [57,58].

Detecting cCMV-associated progressive uSNHL during the first years of life without universal neonatal cCMV screening is a challenge. Clinical protocols for finding these patients needs careful consideration, as they are potential CI candidates [30,59,60,61].

### 4.3. Affected Mechanisms and Auditory Profiles

The aim of this study included describing affected mechanisms in congenital uSNHL. We found that the ABR I/O function for the NE and IE were parallel shifted (Figure 3) and interaural amplitudes were significantly different at higher input levels, in contrast to the pattern previously observed in adults with cochlear hearing losses with recruitment [34,35]. Additionally, no significant interaural difference was found in ABR I-V intervals, indicating no reflection of auditory deprivation or retrocochlear influence on the IEs. The interaural difference in TEOAEs primarily supported the diagnosis of congenital uSNHL, with a NE and an IE from birth. The recordings of DPOAEs confirmed that the TEOAEs recorded here are more sensitive than DPOAEs evoked at 75/75 dB SPL in identifying minor cochlear hearing losses [52]. Not surprisingly, the subjects with DPOAEs at three frequencies or more demonstrated the lowest IE ABRthrs (i.e., 35–40 dB nHL). Similarly, the subjects that lacked DPOAEs also showed some of the highest IE ABRthrs (i.e., ≥60 dB nHL), which adds to research indicating that DPOAE I/O functions in children may be related to their behavior audiometric thresholds [62], although the relationship needs more study.

The parallel shift in ABR I/O functions was supported by the statistically significant effect of ear and stimulus level on waves I, III and V in the linear mixed model. From studies of adults with cochlear hearing loss and loudness recruitment, ABR amplitudes are expected to be significantly lower near the ABRthr in the IE than at the corresponding stimulus level in the NE, with similar amplitudes in the IE and NE at higher ABR stimulus levels of about 70–80 dB HL [34] and 60–80 dB nHL [35]. Thus, the parallel shifted functions suggest a reflection of a lack of recruitment on the group level, and/or a different type of hearing loss than cochlear hearing loss, predominantly affecting outer hair cells [34,35].

The absence of loudness recruitment was also reflected in the ARTs, as a significant interaural difference existed on the group level (Table 1). This agrees with previous study results, e.g., it was found that with higher stimulus levels, where equal loudness was judged between the ears, ARTs did not differ between the ears in adults with mild to moderate unilateral or asymmetrical SNHL and recruitment [63]. Additionally, this finding has been supported in human quinine experiments (induced hearing loss by affecting outer hair cells), where recruitment was reflected in changes in both in the TEOAE I/O function [64], and in ABR I/O function [35], with no change in ARTs, even up to a 46 dB hearing threshold shift [35,64]. It should be noted that parallel shifted NE and IE ABR I/O functions are typical for conductive hearing loss too. However, the interaural ABR latency difference for wave I, III and V was typical for SNHL, i.e., within the level of reproducibility (of about 0.2–0.3 ms) for NEs and IEs with cochlear hearing loss [65,66] (Figure 2 and Appendix A). Further support of no conductive component to the congenital uSNHL can be found in the indistinguishable tympanograms in the NE and IE, and the 10–15 dB interaural difference in ARTs.

ABR I/O functions in combination with ARTs may be useful clinical tools, especially the ABR, as it is already measured as a first or second step by many UNHS programs, and ABR I/O functions have shown to be similar for wave I and V from birth to adulthood [67,68]. The lack of loudness recruitment based on ABR I/O functions and ARTs on group level in children with congenital uSNHL presumably reflect the etiology behind the hearing loss, where, e.g., malformations of the auditory system are more common in unilateral compared to bilateral SNHL [16,17].

The ABR I/O function may add important information to be used in the choice of amplification and compression settings for pediatric patients. Most amplification guidelines for pediatric SNHL recommend the use of validated pediatric-focused prescriptive formulas, i.e., the desired sensation level (DSL) v.5 [69,70] or the national acoustic laboratory non-linear 2 (NAL NL2) [71] that focus on ensuring there is enough compression in the dynamic range to make sounds audible near thresholds and encourage that a wide range of input levels be compressed sufficiently [72,73]. These prescriptive formulas use the hearing thresholds as a function of frequency as a basis and take into account some additional auditory parameters. For loudness, the possibility of choosing between linear gain or wide dynamic range compression (WDRC) exists, and adjustments of the compression threshold can be made, and loudness discomfort levels (LDL) are taken into account for DSL v.5 [69,70]. The LDL is, to our knowledge, the only supra-threshold clinical data that can be applied to modify the prescriptive formulas of amplification. LDLs are not possible to measure in small children, and variability across subjects of all ages is large [74]. Thus, more research is needed into how supra-threshold data that reflect the individual’s reduced dynamic range of hearing could be used in prescriptions to infants and children, e.g., the ABR I/O function studied here, as our groups’ IE vs. NE I/O functions were different than those of adult listeners with cochlear hearing loss [34,35].

### 4.4. Strengths and Limitations

A strength of this diagnostic study is the prospective design of consecutively recruiting all children with congenital uSNHL in Region Stockholm during a two-year period, reducing sample bias. The median 55 dB nHL ABRthr, the 50% males and 65% left IEs (*n* = 20) were similar to the median 50 dB nHL ABRthr, the 56% males and 61% left IEs (*n* = 18) with congenital uSNHL previously screened in the UNHS program in Region Stockholm (>30,000 consecutively screened newborns, 98% coverage rate) [2], indicating a representative sample. Moreover, the median diagnostic age was much lower at 2 months of age, compared to about 4 years of age in etiology/imaging studies using similar subject groups [12,20,21]. Another strength is the detailed descriptions of auditory profiles and that all subjects revealed ABRthrs of ≤20 dB nHL in the NEs at the initial research visit, with healthy outer and middle ears bilaterally. The same audiological test instruments and audiologic test room was used for the measurements. ABRs, ARTs, TEOAEs, DPOAEs and tympanograms were recorded in an automatic and objective manner, and possible remaining between-subject variability due to, e.g., the inadequate placement of earphones or electrodes, was reduced by M.D., M.J. and E.B. performing and analyzing the audiologic measurements. The ABR wave analysis was performed in an objective and blinded manner. K.E. reviewed all MRI scans additional to the first review by one or two experienced neuroradiologists or head-neck radiologists.

Various recruitment steps were undertaken to invite all children born with uSNHL and excluding all children with temporary hearing loss. However, some children may have been missed in the recruitment process, despite our efforts to differentiate conductive from sensorineural components based on the tympanograms, ARTs, otomicroscopy by an experienced otologist, ABR amplitudes and latencies, and our attempts to reschedule follow-up visits to ensure normal hearing once OME was resolved. Based on earlier screening results in Region Stockholm, we expected about ten more participants (Figure 1), with 51,600 newborns screened in January 2019–October 2020 [46] and 0.058% congenital uSNHL prevalence [2], similar to other UNHS programs, e.g., 0.045% [1]. The COVID-19 pandemic may have contributed due to hospital anxiety in the beginning stages of the infection spread in Sweden, as we recruited 3 subjects born during the first 6 months of 2020 when the pandemic arrived in Sweden, compared to 6 subjects per 6 months in 2019. During the last 4 months of the recruitment of children born between July and October 2020 (recruitment stopped 31st of December 2020), after the first Swedish wave of COVID-19 infection, we recruited 5 infants, the expected number.

## 5. Conclusions

In this prospective etiology study of infants with congenital uSNHL, 64% of the MRIs revealed abnormal anatomical structures. In ears with a profound degree of uSNHL the malformation incidence was especially high at 86% (*n* = 6/7), though considerably high also in ears with mild to severe uSNHL at 43% (*n* = 3/7). The most common malformation was an absence of a cochlear nerve found in 50% of the MRI scans, followed by 29% inner ear malformations, of which 21% of all MRIs demonstrated EVA. Thus, MRI is powerful in finding possible causes for congenital uSNHL and may be used to find CI candidates with risks of unilateral to bilateral progression (e.g., ears with EVA). All 20 subjects tested negative for cCMV infection, indicating that it may be relatively uncommon with cCMV infection with very early uSNHL onset, although fluctuations in infection exposure over time should not be overlooked. The parallel shifted IE and NE ABR I/O functions and the interaural ART difference indicate that many children with mild-severe congenital uSNHL may not experience typical loudness recruitment associated with cochlear hearing loss. More research is needed on how supra-threshold measurements, such as the ABR I/O function and the ART, can be used to improve amplification and compression settings in pediatric hearing aids, as poor fittings may contribute to the large variability in hearing aid use and outcomes for children with uSNHL.

## Figures and Tables

**Figure 1 jcm-11-03966-f001:**
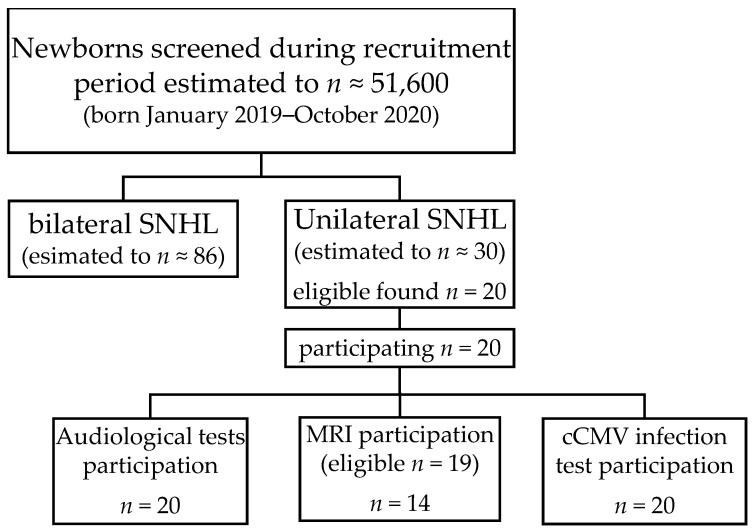
Estimated and included number of participants. Estimated numbers are based on previous universal neonatal hearing-screening results from Region Stockholm ([2], >30,000 screened, 98% coverage rate), and the documented number of newborns during 2019 and 2020 in Region Stockholm [46]. One subject was born with a heart anomaly, which is the reason for 19 subjects eligible for MRI. cCMV = congenital cytomegalovirus; MRI = magnetic resonance imaging; SNHL = sensorineural hearing loss.

**Figure 2 jcm-11-03966-f002:**
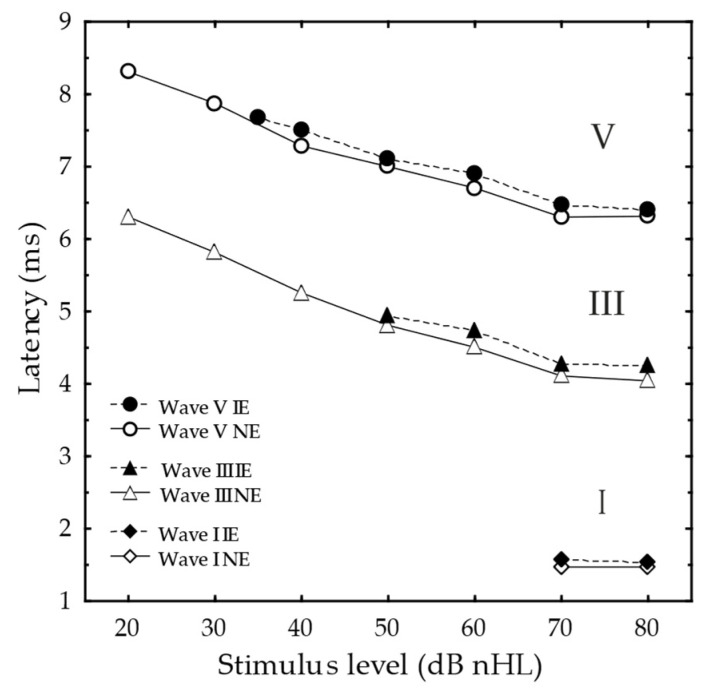
Mean Auditory brainstem response latency as a function of stimulus level (dB nHL) for wave V (circles), wave III (triangles) and wave I (rhombi) for the IE (filled) and the NE (open). The *ns* = 1, 6, 10, 9, 12, and 11 for wave I in the IE for 35, 40, 50, 60, 70, and 80 dB nHL, respectively, whereas the *ns* = 20, 16, 16, 13, 9, 20, and 9 in the NE for 20, 30, 40, 50, 60, 70, and 80 dB nHL, respectively. For wave III the *ns* = 7, 5, 11 and 10 in the IE for 50, 60, 70, and 80 dB nHL, respectively, whereas the *ns* = 11, 13, 14, 11, 9, 20, and 9 in the NE for 20, 30, 40, 50, 60, 70, and 80 dB nHL, respectively. The wave I *ns* = 8 and 7 for the IE and *ns* = 17 and 8 for the NE at 70 and 80 dB nHL, respectively.

**Figure 3 jcm-11-03966-f003:**
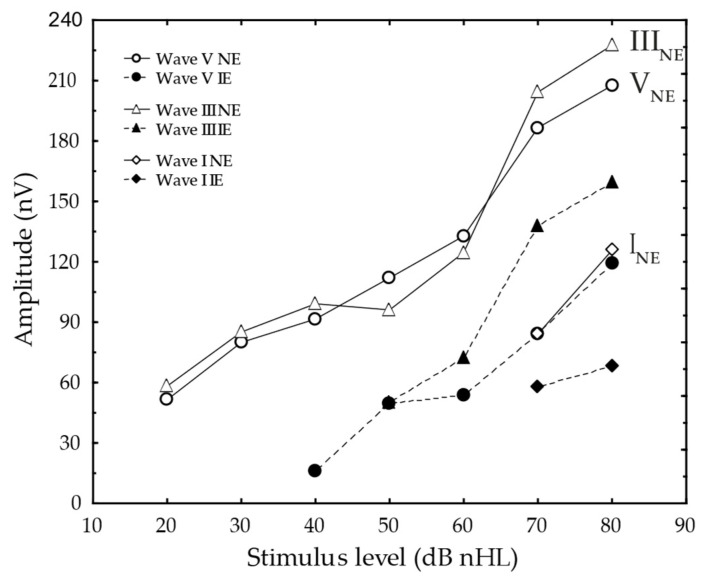
Mean auditory brainstem response amplitude as a function of stimulus level (dB nHL) for wave V (circles), wave III (triangles) and wave I (rhombi) demonstrates parallel shifted I/O functions, when comparing the NE (open) with the IE (filled). The *n*s = 20, 15, 16, 13, 9, 10, and 9 for Wave V in the NE for 20, 30, 40, 50, 60, 70, and 80 dB nHL, respectively, whereas *n*s = 6, 10, 9, 12, and 11 in the IE for 40, 50, 60, 70, and 80 dB nHL, respectively. For wave III the *n*s = 11, 12, 14, 11, 9, 20, and 9 in the NE for 20, 30, 40, 50, 60, 70, and 80 dB nHL, respectively, whereas *n*s = 7, 5, 11, and 10 in the IE for 50, 60, 70, and 80 dB nHL, respectively. The wave I *ns* = 8 and 7 for the IE and *ns* = 17 and 8 for the NE at 70 and 80 dB nHL, respectively.

**Figure 4 jcm-11-03966-f004:**
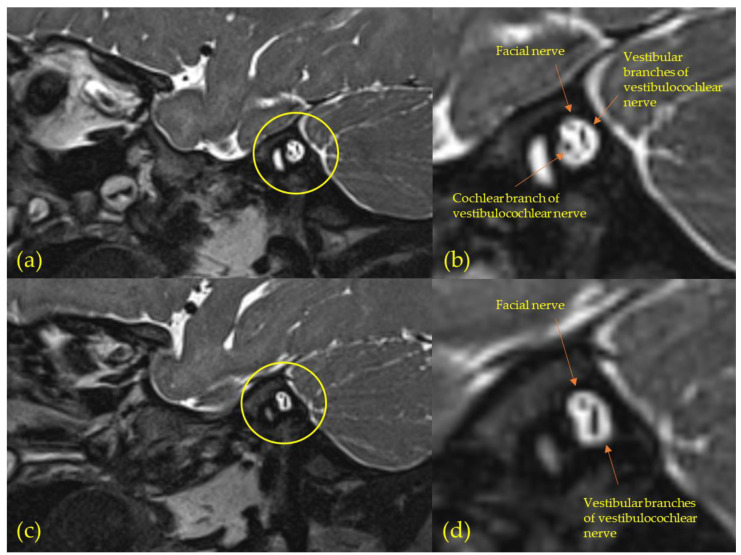
Typical MRI images of the inner auditory canal for subjects with cochlear nerve aplasia (oblique-sagittal view); (**a**,**b**) The normal-hearing ear (NE) of subject 11 (S11) showing a normal cochlear branch of the vestibucochlear nerve; (**c**,**d**) The impaired ear (IE) of S11 showing no visible cochlear nerve branch, indicating aplasia or severe hypoplasia.

**Table 1 jcm-11-03966-t001:** Characteristics of the 20 infants’ impaired ears (IEs), with normal ear (NE) characteristics in parentheses when applicable (all NEs showed ABR thresholds of ≤20 dB nHL and are therefore not in parentheses).

ID	IE	Sex	Age at Diagnosis (mos)	ABR Threshold IE (dB nHL)	ABR Wave I–V (ms)	ART (dB HL)	DPOAE (3 Frequencies or More)	MRI Result
1	L	M	1.8	45	5.07 (4.90)			
2	L	F	2.5	35	4.53 (3.93)		X	N.A.D.
3	L	F	2.1	>90		95 (90)		Hypoplasia cochlea, aplasia/severe hypoplasia auditory nerve, semicircular canal dysplasia and EVA
4	R	F	2.2	40		80 (NT)	X	
5	L	M	2.1	40	4.87 (4.70)	95 (90)	X	N.A.D.
6	L	M	2.0	>90		>105 (75)		Aplasia cochlea, aplasia/severe hypoplasia auditory nerve, labyrinth dysplasia and semicircular canal dysplasia
7	R	F	2.0	45	4.73 (4.73)	80 (75)	X	Bilateral EVA with probable IP II
8	R	F	5.3	>90		95 (85)		Aplasia/severe hypoplasia auditory nerve and hypoplasia inner ear canal
9	L	M	2.2	>90		>105 (85)		
10	L	M	2.7	40	4.94 (5.13)	95 (85)	X	
11	R	M	3.3	>90		>105 (90)		Aplasia/severe hypoplasia auditory nerve and hypoplasia inner ear canal
12	R	F	1.8	>90		>105 (85)		Aplasia/severe hypoplasia auditory nerve and hypoplasia inner ear canal
13	L	F	5.6	>90				N.A.D.
14	L	M	1.7	>90		95 (80)		Aplasia/severe hypoplasia auditory nerve
15	L	M	2.0	40	4.84 (4.80)			N.A.D.
16	L	F	2.3	45	4.90 (4.57)			N.A.D.
17	L	M	2.9	60		100 (75)		Aplasia/severe hypoplasia auditory nerve
18	R	F	3.0	60		100 (80)		Unilateral EVA with probable IP II
19	L	M	4.3	40	4.40 (4.87)			
20	R	F	2.4	50	5.60 (4.83)	85 (75)		

ABR = auditory brainstem response; ART = acoustic reflex threshold, ipsilateral, 1000 Hz; DPOAE = distortion-product otoacoustic emission with SNR > 0 dB; EVA = enlarged vestibular aqueduct; IE = impaired ear; IP II = cochlear incomplete partition type II; L = left; MRI = magnetic resonance imaging; N.A.D. = no anomaly detected; NT = no test; R = right.

## Data Availability

Data are sharable upon request.

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
