# Peer review of "A Prospective Study of Etiology and Auditory Profiles in Infants with Congenital Unilateral Sensorineural Hearing Loss"

_jcm, 2022, doi:10.3390/jcm11143966_

Round 1

Reviewer 1 Report

The authors examine how often and with which diagnostic features a congenital hearing impairment (mild to profound) is associated. The study reads well, but the differences from previously published articles are marginal. In addition, the study population is too ingomogeneous (mild to profound hearing impairment). Such an approach would be interesting, but then a division into subgroups should be made (with larger group sizes). In addition, it is surprising that so few patients present with mild/moderate hearing impairment. Maybe this has to do with the low age of the population (2 months). This circumstance would certainly have to be looked at in more detail.

Reviewer 2 Report

Very interesting paper on congenital unilateral hearing loss in children. Appropriate structure preserved. The abstract corresponds to the topic. Topic presented comprehensively. Conclusions consistent with the purpose of the work. Clear presentation of results and interesting discussion. References to current literature.

Very important topic, knowledge of etiology, allow to implement rapid and proper medical intervention. In the case of children it is extremely important, gives the opportunity for normal auditory and speech development.

Reviewer 3 Report

This manuscript presented an interesting topic using a prospective cross-sectional design to study infants with congenital unilateral sensorineural hearing loss. The manuscript was well-written, and the results addressed several essential findings which help to remind the physicians about the issue of congenital unilateral SNHL. Therefore, I have only a few minor criticisms that must be clarified.

Minor Criticisms:

1.      This study recruited 20 infants for analysis and excluded most potential subjects (48 cases) due to temporary otitis media with effusion (OME). As the OME might resolve within three months, are there any further concerns about no longer waiting and giving up those potential candidates? The revised version may address the authors' reply.

2.      This study defined the pass criteria for TEOAE (SNR³4 dB) but omitted the DPOAE. What is a positive SNR for DPOAE?

3.      Line 402: The authors described that the DPOAE response levels were generally weaker in the IEs compared to the NEs. Do the DPOAE response levels represent the SNR or the signal response only?
